# Similarities and Differences in the Learning Profiles of Adolescents with SLD and SLI in Mathematics—A Preliminary Analysis

**DOI:** 10.3390/brainsci11070850

**Published:** 2021-06-25

**Authors:** Eleni Bonti, Afroditi Kamari, Maria Sofologi, Sofia Giannoglou, Georgia-Nektaria Porfyri, Paraskevi Tatsiopoulou, Georgios Kougioumtzis, Maria Efstratopoulou, Ioannis Diakogiannis

**Affiliations:** 1First Psychiatric Clinic, “Papageorgiou” General Hospital, Ring Road Thessaloniki, N. Efkarpia, School of Medicine, Faculty of Health Sciences, Aristotle University of Thessaloniki, 54603 Thessaloniki, Greece; afrkamari@yahoo.gr (A.K.); sophiegianno@gmail.com (S.G.); geoporfyri@hotmail.fr (G.-N.P.); vivitatsiopoulou@yahoo.gr (P.T.); idiakogiannis@auth.gr (I.D.); 2Department of Education, School of Education, University of Nicosia, Nicosia 2417, Cyprus; 3Laboratory of Psychology, Department of Early Childhood Education, School of Education, University of Ioannina, 45110 Ioannina, Greece; m.sofologi@uoi.gr; 4Institute of Humanities and Social Sciences, University Research Center of Ioannina (URCI), 45110 Ioannina, Greece; 5Department of Turkish and Modern Asian Studies, National and Kapodistrian University of Athens, 10680 Athens, Greece; georgetype@gmail.com; 6Department of Special Education (CEDU), United Arab Emirates University (UAEU), Al Ain 112612, United Arab Emirates; efstratopoulou@gmail.com

**Keywords:** adolescence, SLD, SLI, specific learning disabilities in mathematics, learning profiles, neurodevelopmental disorders

## Abstract

SLI and SLD constitute two independent neurodevelopmental disorders, which frequently cause challenges in the diagnosis process, especially due to their nature. This has caused disagreement among clinicians regarding their recognition as separate or overlapping disorders. The objective of the study was to enlighten the path of valid diagnosis and intervention during adolescence when the two disorders change their manifestation and overlap. Two hundred Greek adolescents (140 boys and 60 girls), 124 already diagnosed with SLD and 76 diagnosed with SLI, 12–16 years old, participated in the study. All participants were assessed in reading, oral and written language and mathematics (mathematical operations and mathematical reasoning) along with IQ testing. In order to determine statistically significant differences, the chi-square test, independent samples *t*-test, odds ratios and their 95 per cent confidence intervals were implemented. The results revealed that the SLI group presented significantly greater difficulties than SLD in their overall cognitive-mental profile and in most language and mathematical measurements (number concept, executive-procedural part of solving operations and mathematical reasoning). The similarity of the two groups was mainly detected in their deficient metacognitive, metalinguistic and metamnemonic strategies. The research concludes that SLD adolescents managed to overcome their difficulties to a significant degree, while adolescents with SLI still struggle with many learning areas.

## 1. Introduction

Specific Language Impairment (SLI) and Specific Learning Disabilities (SLD) constitute two independent neurodevelopmental disorders, which appear to be directly associated, often causing challenges in the diagnosis process. More precisely, differential diagnosis is often a challenging task for clinicians, since the nature and manifestation of the two disorders pose obstacles in deciding whether it is the same language disorder or as two distinct [1,2] but overlapping disorders [3,4]. This overlap may be evident in several symptoms that children with SLD and SLI share, such as problems in reading comprehension, phonological processing, morph syntax or short-term memory deficits and in difficulties with rapid automatic naming [5]. Therefore, many children with SLD manifest language impairments, while it is also common for children with SLI to present SLD symptoms. This suggests that the two disorders pertain to a broader and undivided structure of language disorder, resulting in dyslexia being considered as a milder dimension of language impairment [6].

More specifically, as Catts et al. [3] mentioned, according to the research traditions, the overlap between the two disorders attributed to three axes: firstly, in the presence of the same cognitive deficit, namely phonological processing, the severity of which, shapes the manifestation of the SLD and/or SLI; secondly, while there is the assumption that phonological deficit is common to both disorders, in the case of SLI, it coexists with other cognitive deficits that worsen oral language performance, as opposed to the corresponding SLD populations; the third research tradition explains the overlap revolving around the axis that SLD and SLI are separate disorders which often co-occur or comorbid.

### 1.1. Learning Disabilities in Mathematics

*Specific language impairment (SLI)* refers to the unexpected difficulties of children in effective comprehension and/or expression of language [7,8,9], despite the lack of factors such as mental retardation, neurological deficits, sensory impairment [10] or autism spectrum disorder (ASD) and irrespective of adequate and appropriate environmental conditions [11]. According to epidemiological data, SLI affects approximately 7% of the general population and is considered as the largest category of language impairments [12,13].

Significant warning signs that indicate the existence of SLI are the use of vocabulary below of 15th centile, the use of incomplete sentences in speech and the slow development of language [14]. During adolescence the effect of SLI is reflected on poor academic performance, low levels of mathematical reasoning, weakness in reading comprehension tasks, high risk of functional illiteracy, increased risk of social withdrawal and emotional problems (depression, reduced self-worth, etc.) [15].

However, children with SLI do not exclusively struggle with language problems, but they also present weaknesses in several areas of arithmetic skills, such as number words, verbal counting, written calculations and in the conquest of complex and more developed counting strategies [16]. Working memory is positively associated with early mathematical skills development, specifically about the phonological loop, in which SLI children face limitation in terms of capacity [17,18]. This deficiency severely affects mathematical competency [16]. Language processes consist of factors that are connected to mathematical learning. More specifically, language initially influences the development of mathematical skill, because it leads to the construction of number and quantity concepts [19]. However, the precise underlying mechanism that regulates the language’s role in mathematics is not yet clearly understood. Nonetheless, there is evidence that explains language involvement in mathematics, for example, in the difficulty of decoding arithmetic symbols, as well as in the high risk of failure in increased language requirement mathematical tasks (e.g., word-problems, mathematical reasoning, etc.), and especially in cases where poor visuospatial working memory skills are also present [20]. As regards exact mathematical calculations, language processes are known to significantly contribute to multiplication, but not in subtraction [21]. For children with SLI, the factors that predictively participate in early numeracy skills are grammatical ability, naming speed and phonological awareness [22]. Finally, the literature suggests that early language impairment determines mathematical ability during adolescence [23].

Specific Learning Disabilities (SLD) are among the most common disorders in school-aged children worldwide, with approximately 5–15% prevalence rates, reaching a 3–7% percentage as regards the specific disorder in mathematics [24]. The highest incidence of learning disabilities is manifested in the field of reading for at least 80% of the student population [25], while the comorbidity rates of SLD in reading with SLD in mathematics ranges between 30–70% [26], which is probably explained by the co-occurrence of deficits in math word problems and deficits in reading comprehension [27,28].

According to Munro (2003) [29], students with math difficulties face weakness in processing information, while an important factor in the proper management of information is the degree of its complexity. The basis of cognitive processes required for solving mathematical tasks is initially structured by the numerical magnitude and cardinality understanding, while calculation and fact retrieval prominently contribute to arithmetic ability along with others cognitive functions, such as executive control (attention and decision making) and working memory in particular [30], which is closely associated with the mathematical ability [31,32,33], especially as regards the components of visuospatial memory [34] and phonological processing [35].

Regarding the math word-problem solving, it has been observed that poor performance is due to a deficit in the coordination of numerical and verbal information and to a weakness in inhibiting the unnecessary information during the execution of the task, as a result of a general impairment in the central executive function and/or impairment in working memory [36]. Children with SLD in math lag behind in solving mathematical problems compared to typical development students, due to impairments in visuospatial working memory and due to cognitive obstacles, which significantly inhibit the ability to solve mathematical tasks, such as fact retrieval from memory, basic arithmetic computations and interpretation of word problems [29,37]. However, the study of Passolunghi and Mammarella [38] revealed that children with poor achievement in problem-solving tasks manifested more deficits in spatial, rather than in visual working memory tasks.

During the school years, the weakness and difficulties in mathematics lead to a fear of reoccurrence of failure and to low self-esteem often accompanied by depression, anxiety symptoms and also by a manifestation of aggressive behavior [39]. In addition to the psychological distress, children with learning disabilities also show an increased possibility of social withdrawal and bullying victimization [25,40]. In adolescence, math anxiety usually increases, especially during the transition from primary to secondary education [41,42], which is one of the more stressful periods for school-aged children [43]. In Van Luit and Toll’s study [44], adolescents showed difficulties in naming speed (mainly naming of numbers), which indicate that more effort and time is required to process information during a mathematical task, deficits in the component of planning, in short-term and working memory, whereas in the area of attention, deficits were less frequently detected.

In addition, adolescents have difficulty with clearly writing numbers and correctly placing them to the corresponding columns, calculating money, finding alternative ways to solve a math problem, measuring ingredients, drawing information from charts and maps and understanding the place value [45]. All of these mathematical concepts require the same underlying cognitive processes involved in other mathematical subjects (such as algebra and fractions), most of which are deficient in children with SLD [46]. However, McCaskey, von Aster, O’ Gorman Tuura and Kucian, (2017) in their study [47], claimed that adolescents with SLD, despite their deficits and weaknesses in mathematical skills, manage to effectively process continuous and discrete magnitudes. In addition, the dominant predictor of arithmetic problem-solving performance for children with mathematical learning disabilities is simultaneously processing, as was revealed by Iglesias-Sarmiento, Deaño, Alfonso, and Conde (2017) [48].

In terms of performance in mathematics, low achievement has been observed in 10% of school-aged children and adolescents with SLD [49], which also affects their future daily life as adults [49,50] causing barriers to daily activities that include numeracy practices, as well as employment issues [49]. As 49% of these people often maintain a level of mathematical skills corresponding to that of primary school children [43], they are faced with a risk of unemployment (twice than the rest of the population), failure in various aspects of their life [51] and vulnerability in experiencing social exclusion [52].

### 1.2. Hypothesis

The present study aimed to compare the psycho-educational profiles of two groups of adolescents that had been diagnosed either with SLD or SLI. For this purpose, four hypotheses were formulated.

The first hypothesis was based on our expectation that the adolescents with SLD and SLI would achieve a lower performance in all reading assessment measures irrespective of their diagnosis, as both groups are known to display weaknesses in morphosyntax and phonology [5], leading us to predict that they would perform as ‘*poor*’ readers (Hypothesis 1). More specifically, as literature reveals, there is a consensus in the language impairment profile in children and adolescents with SLI and SLD symptoms, which suggests that the two disorders pertain to a broader and undivided structure of language disorder, resulting in dyslexia being considered as a milder dimension of language impairment [5].

As for the second research hypothesis, we assumed that the SLI group participants would show lower performance in different mathematical assessment tasks, when compared with the SLD group of participants, mainly in mathematical reasoning and operations on account of their weak computational skills and undeveloped counting strategies (Hypothesis 2). Research has shown that verbal working memory is directly involved in different mathematical tasks. As a result, the limitations on working memory capacity are closely aligned with low performances in tasks demanding mathematical competency and numeric cognition [16].

According to the third hypothesis, we expected SLI participants to have a lower performance in all number cognition tasks in comparison to the SLD group (Hypothesis 3). Language has a strong impact on the developmental continuum of mathematical ability since it leads to the construction of number and quantity concepts [20]. Hence, it was expected that language impairment would significantly interfere with mathematical learning and mathematical concepts’ comprehension in the SLI group.

Finally, according to the fourth hypothesis, we predicted that the SLI adolescent group would also present difficulties in instruction understanding of the mathematical problems when compared to the SLD group. Relevant studies have proven that mathematical deficiencies in SLI adolescents are mirrored in difficulties with decoding symbols and in coping with language requirement tasks in mathematics [20].

## 2. Materials and Methods

### 2.1. Participants and Procedure

The purpose of this study was to investigate and to compare the learning profiles of 200 adolescents aged between 12 to 16 years, with a mean age of 13 years and 7 months. Participants consisted of a total of 124 adolescents, who were diagnosed with SLD (91(73.4%) boys and 33(26.6%) girls)) and 76 adolescents diagnosed with SLI, 49 ((64.5%) and 27 (35.5%) girls (Table 1). Both adolescent groups were evaluated in terms of their mental capacity and individual learning areas, such as oral and written language, and mathematical skills (i.e., operation skills and mathematical reasoning abilities), between 2009 and 2014 at the Outpatient State Diagnostic Department for Learning Difficulties (OSDDL) at the First Psychiatric Clinic of “Papageorgiou” General Hospital of Thessaloniki.

The context of the diagnostic procedure involved a psychiatrist, a psychologist and an educational specialist, as provided by DSM–IV [53] and DSM-5 diagnostic criteria [24]. All participants were attending mainstream secondary schools, were native Greek speakers and were referred for evaluation either from their parents’ initiative or following their teachers’ suggestions. It is important to note that the whole sample had no history of neurological disorders or sensory deficits neither had been diagnosed with mental retardation or autism.

### 2.2. Assessment Tools

In Greece, there are several standardized ability/skills or achievement tests, each one of them assessing a particular cognitive or academic area (ex. language, phonological skills, mathematics, etc.). These tests are time consuming, providing information only for a single ability or academic area, or due to their age limits they do not include adolescent students. Moreover, since there is not a commonly accepted assessment battery/tool among the certified Diagnostic Centers, the common practice is that each professional uses his/her own assessment/evaluation tools. Therefore, the assessment tasks used in the present study for the evaluation of the literacy, language and mathematics skills of the participants, were tasks that have been constructed for this purpose [54]. All participants were assessed with the same tasks (evaluating basic—non-curriculum-based academic skills in the areas of literacy, language and mathematics). Each of the skills/tasks was ‘scored’ based on the frequency or the level at which difficulties were detected (0 = none or very rare, 1 = quite often, 2 = very often or systematically).

The assessment tools used were the following:

*IQ measurement***:** the Greek version of WISC III [55] was used to evaluate verbal and non-verbal intelligence.


*Oral reading Skills*


Text reading: The participants were given a three-paragraph text (a simple literary story) and were asked to read it aloud. The examiner recorded the students’ oral reading behavior in terms of their decoding and comprehension abilities [56].Oral decoding skills: Decoding skills were assessed following the ‘miscue analysis’ [57] method of reading modified by Bonti [54], based on the frequency/level of word-by-word reading, ‘dyslexic type’ falsifications (omissions, inversions or reversals, insertions, substitutions), guesses at words, poor pronunciation, poor overall expression, finger-pointing or other behaviors, such as voicing, lip and/or head movements.Comprehension abilities: Participants were asked to answer five comprehension questions in a written form, two of which required simple retrieval of information from the text, while the rest required the ability to either ‘extract’ deeper information implied within the text (i.e., reading ‘between the lines’) or expressing their own understanding of the text. They were also asked to extract the main title for the whole text and subtitles for each paragraph.Study skills abilities: Overall study skills were assessed based on the prevalence (or not) of the following learning characteristics: Low rate of speed, inability to adjust the reading rate, high rate of reading at the expense of accuracy, inability to skim or scan, difficulties locating information, inability or difficulties in extracting a general, appropriate title and subtitles for each of the three paragraphs.Decoding of pseudowords: A list of 20 pseudowords was given to the students and they were asked to read them aloud. Their score was based on their accuracy, speed and decoding abilities.Phonological awareness: The students were given ten oral tasks which assessed their ability to manipulate phonemes, their awareness of phoneme-grapheme relationships, as well as their ability to discriminate between the concepts ‘letter’, ‘word’, ‘syllable’, ‘sentence’ (analysis and synthesis (phonemic segmentation) of letters– syllables containing complex consonant blends, digraphs and other special letter combinations, counting of words within a sentence, or syllables/letters within a word).


*Oral language skills*


The students’ oral language skills were evaluated both through the use of several tasks, but also throughout the whole assessment session (interview, students’ ability to describe their strategies while carrying out several linguistic and non-linguistic relevant tasks). The oral language tasks included the following:Oral expression—narrative skillsLists of opposites and synonymsOral word repetition—auditory memory skills: the students were given five tasks in which they were asked to orally repeat a list of words (3 up to 7 words) with no conceptual relations between them.Oral Sentence repetitionRecognition of verbs, names and adjectives


*Written language skills*


Participants were asked to write a short essay, given a particular subject without a time limit. Their written language skills were assessed taking into account the following tasks:HandwritingSpellingVisual memory skills for linguistic symbols (capital and low case letters in a row): The students were asked to memorize a row of letters (both capital and low case in mixed order), after seeing them for about 15 s and rewrite them.

Their written expression skills (essay) were also evaluated based on: the content (ideas, sufficient vocabulary), the overall expression, the structure and the efficient use of punctuation.


*Mathematics–arithmetic skills*


All participants were asked to solve the same word problem, which required four operations -including two and three-digital numbers- (addition, subtraction, multiplication and division). Therefore, the students’ mathematical skills were evaluated both for their mathematical reasoning ability, as well as for their ability to follow the correct procedures required to carry out the four operations.

Although the students’ overall mathematical skills were evaluated (and scored by the 0, 1, 2 manner) based on the above two basic parameters, more detailed information concerning their mathematical—arithmetic skills were also recorded, based on the observation of their strategic behavior throughout the assessment process. More specifically, whilst carrying out the operations, the examiner recorded the presence or absence of the following skills and/or errors: basic computational skills, direction miscues, use of the ‘traditional’ or a different process (of their own invention), the concept of number, recognition of place value, finger counting, the ability to automatically withdraw multiplications tables from memory, etc.

Their mathematical *reasoning ability/skills* assessment also included the following: Reading and understanding the text in the word story problem, identifying the operations needed to be carried out, (familiarity with mathematics vocabulary and keywords) and the order to be followed, the students’ ability to organize their reasoning and decisions on the steps/procedure they would follow for solving the problem, as well as their ability to orally describe their reasoning.

### 2.3. Data Analysis

Descriptive statistics were used to evaluate data completeness and to characterize the response distributions.

Parametic (independent sample *t*-test) methods were undertaken in order to explore the statistical significance of the observed differences for IQ (total, verbal, practical and subcategories) between adolescents with different diagnosis (SLD, SLI). Parametic methods were chosen because the assumption of normality was not violated. The deviation from the normal distribution was tested using the Kolmogorov–Smirnov test and in all cases the assumption of normality was met (*p* > 0.05).

Furthermore, the chi-square test, odds rations and their 95 percent confidence intervals were used to determine statistical significance differences between adolescents with different diagnosis with respect to different categories of problems.

## 3. Results

Table 2 summarizes the results for the IQ scores using mean and standard deviation. The total IQ, the verbal IQ and the practical IQ mean scores for adolescents with SLD were 100.85 (SD: 11.41), 103.94 (SD: 11.4) and 96.52 (SD: 11.81), respectively, while in adolescents with SLI the scores were 87.71 (SD: 11.17), 84.88 (SD: 11.09) and 93.60 (SD: 14.17). An independent samples *t*-test was used to compare scores of the two groups. The observed difference was statistically significant for the total IQ score (*t*(198) = 7.971, *p* < 0.05) and for the verbal IQ score (*t*(195) = 11.514, *p* < 0.05), while there was not a statistically significant difference for the practical IQ score (*t*(195) = 1.558, *p* > 0.05). Adolescents with SLD had higher total and verbal IQ scores.

Furthermore, the results of these analyses demonstrated a significant difference between the two groups for the sub-scale of Similarities (*t*(193) = 4.120, *p* < 0.05) and sub-scale of Information (*t*(193) = 6.475, *p* < 0.05). In these categories adolescents with SLD had greater scores compared to the adolescents diagnosed with SLI. The chi-square test, odds ratios and their 95 per cent confidence intervals were utilized to determine statistically significant differences between adolescents with SLI and adolescents with SLD in reading, language and mathematical skills.

Table 3 shows the results of the association between text comprehension difficulties and diagnosis. More specifically, adolescents with SLI more frequently exhibited problems related to retrieving simple information questions (χ^2^(1,Ν = 200), *p* < 0.05, OR = 12.667, CI = 5.762–27.848), inferences (χ^2^(1,Ν = 200), *p* < 0.05, OR = 41.379, CI = 18.068–94.762), and giving titles (χ^2^(1,Ν = 200), *p* < 0.05, OR = 22.582, CI = 7.766–65.666), than the SLD group.

The results from the association between oral language difficulties and diagnosis are also presented in Table 3. More specifically, adolescents with SLI more frequently exhibited difficulties with story reproduction (χ^2^(1,Ν = 200), *p* < 0.05, OR = 202.3, CI = 63.467–643), with synonyms/opposites (χ^2^(1,Ν = 200), *p* < 0.05, OR = 37.154, CI = 15.263–90.439), with oral sentence reproduction (χ^2^(1,Ν = 200), *p* < 0.05, OR = 1484.33, CI = 242.3–9093) and with auditory oral word reproduction (χ^2^(1,Ν = 200), *p* < 0.05), than the adolescents with SLD.

In the area of written language skills, the statistical analyses reported only a few statistically significant differences between the two groups. In particular, the SLI adolescents had 61.765 more odds showing poor content compared to SLD adolescents (χ^2^(1,Ν = 200), *p* < 0.05, OR = 61.765, CI = 8.322–458.41). Statistical differences were also found in the poor structure (χ^2^(1,Ν = 200), *p* < 0.05, OR = 7.301, CI = 5.923–57.732) and the poor use of punctuation (χ^2^(1,Ν = 200), *p* < 0.05, OR = 2.296, CI = 1.055–4.994), while difficulties in spelling was a common problem both for SLD and SLI adolescents (χ^2^(1,Ν = 200), *p* > 0.05, OP = 1.698, CI = 0.872–3.306). Concerning difficulties in reading, Table 4 shows the results of the overall reading mechanism evaluation of the two groups, which revealed that most participants of the SLI sample faced weakness in this area of assessment. Moreover, participants with SLI were found to be at a higher risk (9.2 times) of developing a reading difficulty, compared to the group of adolescents with SLD.

In the mathematical skills learning domain (Table 5, statistical analyses revealed that the SLI adolescents had 2.247 more odds, presenting difficulties in operations, compared to the SLD group (χ^2^(1,Ν = 200, *p* < 0.05, OR = 8.863, CI = 4.489–17.500). Besides, the SLI group had a 5.89 higher possibility of developing math difficulties and more specifically, of showing greater incapacities in mathematical operations mainly in the fields of number concepts and the procedural part.

Table 6 presents the results of the mathematical reasoning capacity assessment in the two study groups. In all of the tasks, adolescents with SLI displayed lower performance and were 8.86 times more likely to show difficulties in the mathematical reasoning domain, compared to their peers with SLD.

## 4. Discussion

In our study, the majority of the SLD group did not show significant difficulties with their mathematical reasoning skills (>68%), including their ability to understand the word problem (>72%), to identify the operations needed to be carried out for solving the problem—which involves the ability to identify key words (>67%), to organize their reasoning (>58%), and their ability to verbally describe their thinking and problem-solving strategies in a comprehensive manner (>70%). In line with the above results, regarding the strong correlation of reading with mathematical skills development, findings confirm the first research hypothesis, as both groups showed high levels of difficulty in the reading mechanism and the group comparison revealed that the vast majority (90.8%) of SLI participants faced significant difficulty in this area (1st Hypothesis). Additionally, results confirmed the second hypothesis, as the SLI group participants showed lower performances in mathematical measurements compared to the corresponding SLD group (2nd Hypothesis).

More specifically, most of their ‘mistakes’ or errors in the area of mathematics regarded calculation miscues (>50%), difficulties with following the traditional and correct written procedure for carrying out the operations (>50%) and thus, using ‘their own’ mental procedures (>31%), miscues related to the direction (<25%) and difficulties remembering the multiplication tables ‘by heart’ (>40%). These findings again can be explained by a particular manifestation of the SLD diagnosis per se (ex. Specific learning difficulties in mathematics, dyscalculia) also evidenced in several studies of SLD children [58,59,60] which might include difficulties with number symbols and calculations, although, once again, they may be less obvious during adolescence.

On the other hand, the SLI group also encountered difficulties with carrying out operations (at about the same percentage as the SLD group—about 10% higher), but their main problems occurred in the area of mathematical reasoning skills, as opposed to the SLD group. This finding can be easily comprehended, because mathematical reasoning, apart from the common/everyday grammatical, syntactical, morphological and vocabulary language skills, also presupposes the mastering and knowledge of a discipline or domain-specific language (‘language of math’) [61,62]. In addition, our findings showed that SLI adolescents had greater difficulty with number concepts (69.8%) than the SLD group (29.1%), thus confirming the third hypothesis (3rd Hypothesis) of our study, demonstrating once again, that language plays a crucial role in the construction of number concepts. In the area of mathematical reasoning, according to the results, it was observed that the SLI group scored higher percentages in terms of comprehension of pronunciation (86.8%) than the SLD group (27.4%). This finding is in line with the fourth research hypothesis (4th Hypothesis) and reflects the SLI group’s weakness in dealing with language demanding mathematical tasks and with decoding arithmetic symbols.

In their research, Ehren, Murza and Malani, 2012 [63] Faggella-Luby et al., 2012 [64], have also stressed the importance of how comprehensive language impairments may prohibit language processing, even in non-solely linguistic academic areas, such as social studies, science, mathematical story problems, etc. SLI adolescents presented an overall lower (albeit within the normal levels) IQ score (total and verbal) compared to the SLD group. According to Alloway, Tewolde, Skipper, and Hijar, (2017) [65], who conducted a study in SLD and SLI children the nonverbal IQ scores were predictively associated with math performance.

Our finding was somehow expected since it agrees with the actual diagnostic criteria of the SLI population. A challenging conceptualization, deriving from this finding, especially in the particular age group (adolescents), was that the ‘ostensibly low’ total IQ score witnessed in most of the SLI adolescents could be a possible ‘plasmatic’ reflection of the SLI child’s ongoing—throughout the school years—struggle with the various academic tasks, due to their ‘problematic’ language skills, rather than vice versa.

Catts et al. (2001) [66]; Olivier et al. (2000) [67], have also argued that language deficits interfere with metalinguistic awareness, problems organizing skills and analyzing information effectively and efficiently. Mathematical reasoning requires all the above abilities, thus explaining why the SLI group experienced significant difficulties in this area. Finally, as research suggests, a common characteristic/deficit of all SLD and SLI students is the lack of metacognitive/metalinguistic and mnemonic strategies [68], which also interfere with the overall process of learning. Once again, based on our findings the SLI adolescents seemed to lack these strategies to a greater extent, compared to the SLD adolescents. Therefore, these strategies presuppose a sufficient general language development, a fact that, once again, points out the severe interference of the deficient language skills, even in the learning and metamnemonic strategies.

Research on typical development students at the end of middle school showed that math self-concept, task persistence and reading comprehension have a beneficial influence on problem-solving and on supplemental learning strategies based on metacognitive awareness and that reading comprehension efficacy could further develop mathematical skills [69]. The results on general learning/academic skills, as have been assessed by the tasks used in this study, even in the ‘strategic’ abilities, showed that the SLD adolescent population has probably outgrown their difficulties, as opposed to SLI adolescent population.

The innovative characteristic of the present study is that the learning skills of students with SLI and SLD examined, focused on the adolescent age (or late school years), which turned out to be the most “appropriate” (albeit less studied). This factor differentiates our findings from other relevant studies, which, even though investigated the same learning skills, their samples consisted of much younger children with SLD and SLI, mainly in the first grades of typical education.

## 5. Conclusions

The current study investigated the cognitive/clinical profiles of two groups of adolescents, already diagnosed with SLD or SLI. The results revealed that adolescents with SLI presented significantly greater difficulties than adolescents with SLD, both in their overall cognitive-mental profile and in most language areas, as well as in the field of mathematics (number concept, executive-procedural skills for solving operations and mathematical reasoning). In particular, the level of ‘errors’ of adolescents with SLI in the area of mathematical operations was higher and presented a minor difference compared to those of the SLD group. Their main difficulty exists in all parameters of mathematical reasoning, which presupposes a basic knowledge of grammar, syntax, morphological structures of language, efficient vocabulary and special language of mathematics. The similarity of the two groups was based on the deficient metacognitive, metalinguistic and metamnemonic strategies, which require adequate language development.

Consequently, the research revealed that the SLD adolescents managed to overcome most of their difficulties to a significant degree, while adolescents with SLI, were still struggling. The main finding was that the SLI group still manifested weaknesses in all academic areas, while the SLD participants only in specific domains. The evolutionary course of learning disabilities seems to be in favor of the ‘unmixed’ learning profile of the SLD group (i.e., symptoms are reduced overtime or limited in specific areas), in contrast to that of children with SLI, whose learning profiles are more complicated.

The research leads us to the conclusion that SLI is a highly complex disorder, which, especially during adolescence, manifests itself in the form of ‘generalized learning difficulties’, evident in all major learning areas, while at the same time, it can lead to a ‘fictional’ image of a low mental level. The above effect can lead to an inaccurate diagnosis (e.g., SLD instead of SLI) and possibly to an inappropriate intervention, while the real cause of the adolescent’s learning difficulties is their ongoing struggle with language difficulties. This is due to the fact that, between the two disorders of SLD and SLI, there are common characteristics and overlaps, often creating confusion among clinicians.

The findings of the present study could be utilized in the future, to better clarify the cognitive profiles of the two respective groups, particularly in the challenging learning area of mathematics. This study will, hopefully, contribute to the future development of better interventions and more appropriate provision of educational support that will better meet the challenging learning needs of the two populations.

## 6. Limitations of the Study and Future Research

In the context of the present study, some limitations and future research recommendations could be considered to further enlighten the clinical profile differences between the two groups. For instance, the implementation of neuropsychological tests at the evaluation stage, particularly in the domain of executive functioning, could be an interesting area of future inquiry, in conjunction with learning assessment, to clarify the neural brain network of these populations. Furthermore, it would be useful to include additional parameters in the investigation of the adolescents’ profiles, such as gender and socioeconomic status, to better investigate how profiles are shaped according to these demographic factors. Moreover, larger samples of adolescent or adult participants could be used in future studies, to allow better levels of generalizability of results based on larger numbers of participants across different age groups.

In addition, further investigation is essential, in relation to the cognitive factors of math anxiety in students (children and/or adolescents) with SLD–SLI, regarding the elaboration of the way the component of math anxiety operates and regarding the degree of its influence on academic performance. Finally, the overall findings of this study suggest the need for including metacognitive learning strategies in mathematics teaching and intervention to reinforce the mathematical skills of secondary education students with SLI and SLD.

## Figures and Tables

**Table 1 brainsci-11-00850-t001:** Demographic characteristics.

	N	Age		Gender	
		M.O	SD	Boys	Girls
SLD SLI	124	13.7	2.33	91 (73.4%)	33 (26.6%)
	76	13.7	2.33	49 (64.5%)	27 (35.5%)

**Table 2 brainsci-11-00850-t002:** Descriptive statistics for IQ with respect to diagnosis and significance of *t*-test.

	SLD		SLI		
	Mean	Standard Deviation	Mean	Standard Deviation	*p*
Total IQ	100.85	11.41	87.71	11.17	0.005
Verbal IQ	103.94	11.40	84.88	11.09	0.005
Practical IQ	96.52	11.81	93.60	14.17	0.121
Information	9.98	2.71	7.39	2.71	0.005
Similarities	11.52	2.62	8.35	2.37	0.005
Vocabulary	7.67	2.73	8.00	2.96	0.428
Filling Images	9.27	2.88	9.01	2.94	0.546
Cubes	10.02	2.77	9.20	2.83	0.051
Object Assembly	10.08	2.78	9.50	2.89	0.164

**Table 3 brainsci-11-00850-t003:** Results of chi-square test and odds ratio for text comprehension, oral and written language, decoding difficulties with respect to diagnosis.

		SLD (%)	SLI (%)	*p*	*^1^ OR (95% CI *^2^)
**Text Comprehension**					
Difficulties in retrieving simple information questions	no yes	91.4 8.1	47.4 52.6	0.005	12.667 (5.762–27.848)
Differences in inferences	no yes	89.5 10.5	17.9 82.9	0.005	41.379 (18.068–94.762)
Difficulties in giving titles	no yes	55.6 44.4	5.3 94.7	0.005	22.582 (7.766–65.666)
**Oral language**					
Difficulties in story reproduction	no yes	96.0 4.0	10.5 89.5	0.005	202.3 (63.467–643.0)
Difficulties in synonyms/opposites	no yes	79.0 21.0	9.2 90.8	0.005	37.154 (15.263–90.439)
Difficulties in auditory oral word reproduction (DTLA-2)	no yes	100.0 0.0	61.8 38.2	0.005	**
**Written language**					
Poor handwriting	no yes	11.3 88.7	0.0 100.0	0.005	-
Difficulties in spelling	no yes	80.6 19.4	70.1 28.9	0.118	1.698 (0.872–3.306)
Poor content	no yes	45.1 54.8	1.3 98.7	0.005	61.765 (8.322–458.41)
Poor Structure	no yes	8.9 91.1	1.3 98.7	0.028	7.301(5.923–57.732)
Poor use of punctuation	no yes	25.8 74.2	13.2 86.8	0.033	2.296(1.055–4.994)
**Decoding Difficulties**					
Substitutions	no yes	56.5 43.5	42.5 57.9	0.049	1.782 (1.001–3.175)
Syllabic reading	no yes	80.6 19.4	71.1 28.9	0.118	1.698 (0.872–3.306)
Line skipping	no yes	92.7 7.3	72.4 27.6	0.005	4.879 (2.097–11.351)
Finger pointing	no yes	79.8 20.2	69.7 30.3	0.104	1.718 (0.891–3.316)
Hesitations	no yes	29.0 71.0	6.6 93.4	0.005	5.809 (2.167–15.576)
Repetitions of syllables, words, phrases	no yes	69.5 31.5	40.8 59.2	0.005	3.164 (1.747–5.731)
Non-acknowledgement of punctuation	no yes	67.7 32.3	19.7 80.3	0.005	8.540 (4.331–16.838)
Difficulties in decoding pseudowords	no yes	57.3 42.7	47.4 52.6	0.173	1.488 (0.839–2.642)

*^1^ OR = Odds Ratio; *^2^ CI = Confidence Interval; ** Cannot be calculated because the relative frequency for SLD adolescents in category No is 0.

**Table 4 brainsci-11-00850-t004:** Reading difficulties.

		SLD (%)	SLI (%)
Difficulty in the reading mechanism	no	48.4	9.2
	yes	51.6	90.8

**Table 5 brainsci-11-00850-t005:** Results for mathematical operations and reasoning with respect to diagnosis.

		SLD (%)	SLΙ (%)	*p*
**Mathematical Operations**				
Difficulty in operations	no yes	57.3 43.7	42.1 57.9	0.037
Mistakes in computations	no few many	47.6 34.7 17.7	40.8 40.1 13.2	0.264
Direction mistakes	no few many	74.2 25.0 0.8	75.0 25.0	0.735
Mistakes in the procedural/executive part	no few many	52.4 25.0 22.6	32.9 43.4 23.7	0.011
Difficulty in concepts about number	no low high	71.0 21.0 8.1	30.3 31.6 38.2	0.005
Weakness in memorizing multiplication tables	no low high	56.5 34.7 8.9	57.9 36.8 5.3	0.639
Use of problem-solving skills	no yes	68.5 31.5	76.3 23.7	0.238
**Reasoning Capacity**				
Difficulty in mathematical reasoning	no yes	68.5 31.5	19.7 80.3	0.005
Difficulty understanding pronunciation	no low high	72.6 24.2 3.2	13.2 50.0 36.8	0.005
Difficulty organizing key-words	no low high	67.7 15.3 16.9	25.0 42.1 32.9	0.005
Difficulty organizing reasoning	no low high	58.9 18.5 22.6	15.8 22.4 61.8	0.005
Difficulty describing steps	no low high	70.7 25.2 4.1	2.6 50.0 47.4	0.005

**Table 6 brainsci-11-00850-t006:** Results of chi-square test and odds ratio for operations difficulties with respect to diagnosis.

		SLD (%)	SLI (%)	*p*	OR (95% CI)
Difficulties in operations	no yes	52.4 47.6	32.9 61.7	0.007	2.247 (1.240–4.072)
Difficulties in mathematical reasoning	no yes	68.5 31.5	19.7 80.3	0.005	8.863 (4.489–17.500)

## Data Availability

The data presented in this study are available on request from the corresponding author due to privacy issues. The data are not publicly available due to privacy.

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
