# Peer review of "Similarities and Differences in the Learning Profiles of Adolescents with SLD and SLI in Mathematics—A Preliminary Analysis"

_brainsci, 2021, doi:10.3390/brainsci11070850_

Round 1

Reviewer 1 Report

The paper presents a study on adolescents that presented difficulties in mathematical concepts and skills for problem solving and mathematical reasoning. The findings reveal that such difficulties are greater in adolescents with Specific Language Impairment (SLI) than in those with Specific Learning Disabilities (SLD). The research is appropriately designed and methods are clearly explained and, up to this reviewer's knowledge, they are valid for the research goal and to test the stated hypotheses.

The findings are interesting and aligned with a present issue dealing with the relationship between language skills, mathematical skills and other skills. Since mathematical deficiencies in SLI adolescents are mirrored in difficulties with decoding symbols, the research posed in this paper is reasonably well founded. Findings are comprehensible because mathematical abilities involves the use of a specific language of Maths --- which is being subject to a current controversy on the need for its evolution (see e.g. C. Wolfram's "The Math(s) Fix: an Education Blueprint for the AI age" [1]).

The research is aligned with recent studies such as Ivanova et al.'s [2], who show that “Understanding computer code seems to be its own thing. It’s not the same as language, and it’s not the same as math and logic”. By researching SLI vs SLD adolescent populations, this article can shed some light on the investigation of the mutual influence of language skills/impairments vs mathematical skills/impairments and other learning skills/impairments. Results would be very relevant to (in the authors' words) "contribute to the future development of better interventions and more appropriate provision of educational support" and thus scientifically analyze whether Wolfram's intent on changing the way of learning Maths is worth it, whether improving language skills have to be more relevant in it, or if Ivanova et al.'s neuroimaging investigations can influence such decisions. Certainly, neuropsychological tests must be implemented as a future inquiry (as authors acknowledge).

A final remark has to do with the use across the text of the term "computational skills" (sec. 1.2; sec. 2-Mathematics-aritmetics skills; sec. 4). It refers to the capability of making computations (by oneself, without a computer support). But it can be confounded with "computational abilities/thinking", which are named after the capabilities needed to solving problems with the help of an automated processor (i.e. computer). See Wolfram's book or Denning & Tedre's book [3] on the topic. Therefore I recommend a clarification of the term "computational skills" or changing it by another more linked to "manual" computation abilities.

[1] https://www.wolfram-media.com/products/the-maths-fix.html

[2] https://doi.org/10.7554/eLife.58906

[3] https://mitpress.mit.edu/books/computational-thinking

Author Response

Response to Reviewer 1 Comments

Point 1. The paper presents a study on adolescents that presented difficulties in mathematical concepts and skills for problem solving and mathematical reasoning. The findings reveal that such difficulties are greater in adolescents with Specific Language Impairments (SLI) than in those with Specific Learning Disabilities (SLD). The research is appropriately designed and methods are clearly explained up and, up this reviewer’s knowledge, they are valid for the research goal and to test the stated hypothesis.

The findings are interesting and aligned with a present issue dealing with the relationship between language skills, mathematical skills and other skills. Since mathematical deficiencies in SLI adolescents are mirrored in difficulties with decoding symbols, the research posed in this paper is reasonably well founded. Findings are comprehensible because mathematical abilities involve the use of a specific language of Math, which is being subject to a current controversy on the need for its evolution (see e.g. C. Wolfram’s “The Math(s) Fix: an Education Blueprint for the Al age” [1]).

The research is aligned with recent studies such as Ivanova et al.’s [2], who show that “Understanding computer code seems to be its own thing. It’s not the same as the language, and it’s no the same as the math and logic”. By researching SLI vs SLD adolescent populations, this article can shed some light on the investigation of the mutual influence of language skills/impairments vs mathematical skills/impairments. Results would be very relevant to (in the authors’ words) “contribute to the future development of better interventions and more appropriate provision of educational support” and thus scientifically analyze whether Wolfram’s intent on changing the way of learning Math’s is worth it, whether improving language skills have to be more relevant in it, or if Ivanova et al. ‘s neuroimaging investigations can influence such decisions. Certainly, neuropsychological tests must be implemented as a future inquiry (as authors acknowledge).

A final remark has to do with the use across the text of the term “computational skills” (sec.1.2; sec. 2-Mathematics-arithmetic skills; sec.4). It refers to the capability of making computations (by oneself, without a computer support). But it can be confounded with “computational abilities-thinking”, which are named after the capabilities needed to solving problems with the help od an automated processor (i.e. computer). See Wolfram’s book or Dennis & Tedre’s book [3] on the topic. Therefore, I recommend a clarification of the term “computational skills” or changing it by another more linked to “manual” computation abilities.

Response 1: We would like to thank the first author for the detailed comments and suggestions for the manuscript and also, for his/her significant contribution in the review of the present article. We used the term “computational skills” as it is found widely in literature, in order to be determined this type of skill or ability.

Mabbott, D.J.; Bisanz, J. (2008). Computational Skills, Working Memory, and Conceptual Knowledge in Older Children With Mathematics Learning Disabilities. Journal of Learning Disabilities, 41(1), 15–28. doi:10.1177/0022219407311003 

Soares, N.; Evans, T.; Patel, D. (2018). Specific learning disabilities in mathematics: A comprehensive review. Translational Pediatrics, 7(1): 48-62. doi:  10.21037/tp.2017.08.03

Hecht, S.A.; Torgesen, J.K.; Wagner, R.K.; Rashotte, C.A.  The relations between phonological processing abilities and emerging individual differences in mathematical computation skills: A longitudinal study from second to fifth grades. Journal of Experimental Child Psychology, 2001, Volume, 79, 192–227, DOI:10.1006/jecp.2000.2586.

Witzel, B.S. (2005). Using cra to teach algebra to students with math difficulties in inclusive settings. Learning Disabilities: A Contemporary Journal 3, 49-60.

Reviewer 2 Report

At the most fundamental level, this paper tries to do too many things. As such, the narrative is dense, difficult to follow and, more importantly, does not clearly convey the most salient message of the title of the research paper. There are problems with the methodology and statistical analyses. Finally, problems with the clarity of written expression and academic writing prevent me from recommending this paper for publication, in its current form.

Fundamentals

At the most basic level, there problems with how the two behavioral disorders are defined and discussed. For example, I am not familiar with the term “’autonomous’ neurodevelopmental disorder” (lines 21 and 41). How can a disorder be ‘autonomous’ and also ‘overlapping’ (line 23)?

Line 57: “In the case of SLI, [phonological deficits] coexists with other cognitive deficits that worsen oral language performance”.

This is a fundamental misrepresentation of the term “Specifical Language Impairment”. By definition, SLI is a challenge in acquiring language in the absence of frank cognitive deficits.

Line 54: It is questionable as to whether phonological processing should be considered a “cognitive deficit”. As above, SLI is a challenge in acquiring language in the absence of frank cognitive deficits. Certainly, individuals with SLI show deficits in phonological processing – this is one of the processing deficits associated with the disorder – but, again, the use of ‘cognitive’ is questionable.

Introduction to the paper

Line 62-65: For me, the connection between low achievement in mathematics and daily living is not obvious. Consider rearranging this sentence/paragraph.

Line 96-105: The discussion of neuroanatomy seems tangential to the paper, which focusses on behavioral measures of language, learning, and mathematics. I’m not sure how this section is relevant, it doesn’t add anything as is.

Lines 192-218: Consider laying out the hypotheses as bullet points with sub-headers, which will make these easier to differentiate.

Methodology

Demographic info: 12-16 years is a large age range, especially given the extent of learning that happens in these later school years. Table 1 shows 2.33 SD across groups in age, which is not insignificant. This is not accounted for in analyses, nor is it directly discussed as it relates to the study’s findings.

Results

A large number of measures of oral and written language, as well as mathematical assessments, are administered across the two sample groups. Given that one of the premises of this paper is to assess the association of these skills, it seems odd that (as far as I can tell), there is no attempt to correct for multiple testing across, or at correlational analyses.

Lines 346-349: Given the difference between total IQ and verbal IQ scores in the SLI vs. SLD group, how can you be sure the study’s findings are not driven by verbal IQ scores (given that problems with language are the hallmark of SLI)?

Line 528: Given that information on gender was available, why was this not examined as statistically significant, or included as a co-variate in analyses?

Clarity of expression and academic writing

  1. There are a number of problems with written expression and grammar that are related to English as a second language. These mean the paper is more difficult to follow.

  1. There are also a number of problems with taxonomy which, I’m guessing, are also related to English as a second language e.g., “the phonological deficit” (line 57) is typically referred to without the determiner e.g., “phonological deficit” or "phonological deficits".

Author Response

Response to Reviewer 2 Comments

Point 2: “At the most basic level, there problems with the two behavioral disorders are defined and discussed. For example, I am not familiar with the term ‘autonomous’ neurodevelopmental disorder (lines 21 and 41). How can a disorder be ‘autonomous’ and also ‘overlapping’ (line 23)?”

Response 1: SLI and SLD are two distinct developmental language disorders; SLI primarily represented by difficulties in semantics, syntax, and discourse, and SLD is characterized by problems in reading, writing and/or mathematics. However, recent findings suggest there may be a closer association between these developmental language disorders. Children with SLD have been shown to have early deficits in semantics and syntax (Gallagher, Frith, & Snowling, 2000; P. Lyytinen, Poikkeus, Laakso, Eklund, & Lyytinen, 2001; Scarborough, 1990, 1991; Snowling, Gallagher, & Frith, 2003), and children with SLI have often been noted to have phonological processing deficits and subsequent problems in word recognition (Catts, 1993; Snowling, Bishop, & Stothard, 2000). These findings have led some to conclude that SLD and SLI represent variants of the same developmental language disorder (Kamhi & Catts, 1986; Tallal, Allard, Miller, & Curtiss, 1997). We used the term “autonomous”, in order to refer to the two neurodevelopmental disorders, as separate (distinct) disorders. Thank you for your suggestion and to prevent possible confusion we changed the adjective “autonomous” and  replaced it as “independent”.

Point 2: Line 57: “In the case of SLI, [phonological deficits] coexists with other cognitive deficits that worsen oral language performance”. This is fundamental misrepresentation of the term “Specific Language Impairment”. By definition, SLI is a challenge in acquiring language in the absence of frank cognitive deficits.”

Response 2: Thank you for offering the opportunity to clarify that language impairments are due to cognitive deficits located in the working memory capacity (magic number, Miller, 7/ + - 2). Because of the low memory capacity, individuals with SLI cannot retain a large number of words (Gathercole & Baddeley, 1995; Baddeley & Hitch, 2000; Alloway, 2007; Baddeley, 2017).

Point 3: Line 54: “It is questionable as to whether phonological processing should be considered a ‘cognitive deficit’. As above, SLI is a challenge in acquiring language in the absence of frank cognitive deficits. Certainly, individuals with SLI show deficits in phonological processing- this is one of the processing deficits associated with the disorder- but again, the use of ‘cognitive is questionable”

Response 3: Thank you very your comment, this point was clarified in the previous response.

Point 4: Line 62-65: “For me, the connection between low achievement in mathematics and daily living is not obvious. Consider rearranging this sentence/paragraph.”

Response 4: Your recommendation was very helpful, thank you again. We made the change concerning this sentence.

Point 5: Line 96-105: The discussion of neuroanatomy seems tangential to the paper, which focuses on behavioral measures of language, learning and mathematics. I’m not sure how this section is relevant, it doesn’t add anything as is.

Response 5: Thank you for the insightful comment, the specific part of the text is bounded to show the brain areas that are responsible for mathematical development and how this development is formed from childhood to adolescence. This part actually reveals the fact that mathematical ability is associates closely with cognitive processes, which occur in specific brain networks.

Point 6: Methodology. Demographic info: 12-16 years is a large age range, especially given the extend of learning that happens in these later school years. Table 1 shows 2.33 SD across groups in age, which is not insignificant. This is not accounted for in analyses, nor is directly discussed as it relates to the study’s findings.

Response 6: Thank you for remarking on this point, however, we would like to clarify that according to the literature a deliberated attempt has been made to minimize reference to age as a criterion for categorization of learners. More specifically, research on lifespan development shows that chronological age per se isn’t the only predictor of learning ability (Santrock, 2006; Vander Zanden, Crandell, & Crandell, 2007; Whitener, Cox, & Maglich, 1998). Personality also plays an important role in the prediction of academic achievement because it affects student motivation and behavior in work situations (e.g., Paunonen & Ashton, 2013; Richardson, Abraham, & Bond, 2012) and in addition, psychological maturation in adolescence is another significant factor (Morales-Vives, Camps, & Dueñas, 2019). However, your remark is useful to be taken into account in the next stage of the research, because the present study is a preliminary analysis of the collected data.

Point 7: Results. A large number of measures of oral and written language, as well as mathematical assessments, are administrated across the two sample groups. Given that one of the premises of this paper is to assess the association of these skills, it seems odd that (as far as I can tell), there is no attempt to correct for multiple testing across, or at correlational analyses

Response 7: This was a preliminary analysis, in the next stage of the investigation will be conducted a correlational analysis. Thank you for point out this field of methodology, we have added to the title of the paper that the study concerns a preliminary analysis.

Point 8: Results. Given the difference between total IQ and verbal IQ scores in the SLI vs SLD group, how can you be sure the study’s findings are not driven by verbal IQ scores (given that problems with language are the hallmark of SLI?

Response 8: Thank you for your comment, because offers the opportunity to clarify that our findings are aligned with the results of previous studies (Ehren, Murza & Malani, 2012; Faggella-Luby et al., 2012) which revealed that SLI adolescents presented an overall lower (albeit within the normal levels) IQ score (total and verbal) compared to the SLD group. According to Alloway, Tewolde, Skipper, & Hijar, (2017) study, the nonverbal IQ scores were predictively associated with math performance. Our finding was somehow expected since it agrees with the actual diagnostic criteria of the SLI population.

Point 9: Results. Given that the information on gender was available, why was this not examined as statistically significant, or as a co-variate in analyses?

Response 9: This was a preliminary analysis, in the next stage of the investigation will be conducted a correlational analysis. Thank you for pointing out this field of methodology, we have added to the title of the paper that the study concerns a preliminary analysis. However, your observation is useful and as we have mentioned in the limitations and the future research recommendations the demographic factors could offer interesting findings in a later stage of the research.